# More Impaired Dynamic Ventilatory Muscle Oxygenation in Congestive Heart Failure than in Chronic Obstructive Pulmonary Disease

**DOI:** 10.3390/jcm8101641

**Published:** 2019-10-07

**Authors:** Ming-Lung Chuang, I-Feng Lin, Meng-Jer Hsieh

**Affiliations:** 1Division of Pulmonary Medicine and Department of Internal Medicine, Chung Shan Medical University Hospital, Taichung 40201, Taiwan; 2School of Medicine, Chung Shan Medical University, Taichung 40201, Taiwan; 3Institute and Department of Public Health, National Yang Ming University, Taipei 11221, Taiwan; iflin@ym.edu.tw; 4Department of Pulmonary and Critical Care Medicine, Chiayi Chang-Gung Memorial Hospital, Chang-Gung Medical Foundation, Chiayi 61363, Taiwan; mengjer@yahoo.com; 5Department of Respiratory Therapy, Chang Gung University, Taoyuan 33302, Taiwan

**Keywords:** near-infrared spectroscopy, frequency domain, heart failure, chronic obstructive pulmonary disease, maximal exercise, vastus lateralis muscle, serratus anterior muscle, cardiopulmonary exercise testing

## Abstract

Patients with chronic obstructive pulmonary disease (COPD) and congestive heart failure (CHF) often have dyspnea. Despite differences in primary organ derangement and similarities in secondary skeletal muscle changes, both patient groups have prominent functional impairment. With similar daily exercise performance in patients with CHF and COPD, we hypothesized that patients with CHF would have worse ventilatory muscle oxygenation than patients with COPD. This study aimed to compare differences in tissue oxygenation and blood capacity between ventilatory muscles and leg muscles and between the two patient groups. Demographic data, lung function, and maximal cardiopulmonary exercise tests were performed in 134 subjects without acute illnesses. Muscle oxygenation and blood capacity were measured using frequency-domain near-infrared spectroscopy (fd-NIRS). We enrolled normal subjects and patients with COPD and CHF. The two patient groups were matched by oxygen-cost diagram scores, New York Heart Association functional classification scores, and modified Medical Research Council scores. COPD was defined as forced expired volume in one second and forced expired vital capacity ratio ≤0.7. CHF was defined as stable heart failure with an ejection fraction ≤49%. The healthy subjects were defined as those with no obvious history of chronic disease. Age, body mass index, cigarette consumption, lung function, and exercise capacity were different across the three groups. Muscle oxygenation and blood capacity were adjusted accordingly. Leg muscles had higher deoxygenation (HHb) and oxygenation (HbO_2_) and lower oxygen saturation (S_m_O_2_) than ventilatory muscles in all participants. The S_m_O_2_ of leg muscles was lower than that of ventilatory muscles because S_m_O_2_ was calculated as HbO_2_/(HHb+HbO_2_), and the HHb of leg muscles was relatively higher than the HbO_2_ of leg muscles. The healthy subjects had higher S_m_O_2_, the patients with COPD had higher HHb, and the patients with CHF had lower HbO_2_ in both muscle groups throughout the tests. The patients with CHF had lower S_m_O_2_ of ventilatory muscles than the patients with COPD at peak exercise (*p* < 0.01). We conclud that fd-NIRS can be used to discriminate tissue oxygenation of different musculatures and disease entities. More studies on interventions on ventilatory muscle oxygenation in patients with CHF and COPD are warranted.

## 1. Introduction

Exercise intolerance and exertional dyspnea are often encountered in patients with chronic obstructive pulmonary disease (COPD) and cardiovascular diseases, and especially in those with congestive heart failure (CHF) [1]. The symptoms are primarily caused by heart and lung pathological changes that lead to physiological limitations, and secondarily by ventilatory and locomotor muscle weakness due to myopathy [1,2,3,4]. Causes of myopathy include hypoxia, oxidative stress, medication, nutritional depletion, systemic inflammation, disuse and atrophy (sarcopenia). However, functional impairment is more prominent than structural derangement. It has been hypothesized that functional impairment is further impaired by exercise due to failure of perfusion to meet the need of increased metabolism. Patients with CHF have primarily reduced central cardiovascular function, whereas patients with COPD have secondarily reduced central cardiovascular function caused by dynamic hyperinflation [5]. Moreover, the locomotor muscles are expected to be more impaired than the inspiratory muscles (diaphragm), as inspiratory muscles are more resistant to fatigue with a higher proportion of type I fibers, despite also being impaired in strength, whereas the quadriceps have a lower proportion of type I fibers [6,7,8]. Even though the mechanisms of myopathy, muscle morphology, muscle fiber type distribution and shifting, and muscle metabolism in the locomotor muscles are similar in patients with COPD and CHF [1,9], exercise-induced hypoxemia, if it occurs, is expected to aggravate hypoxic stress in patients with COPD, as the patterns of muscle cytochrome oxidase gene activation are altered in these patients [8]. In addition, hypoxemia may cause an increase in myostatin protein expression, which is related to muscle atrophy [10], and patients with COPD can experience vascular dysfunction with a higher extraction of muscle oxygen during exercise even in the early stage [11]. Therefore, it is not clear whether there are differences in the functions of locomotor and ventilatory muscles between patients with COPD and CHF who have similar daily exercise performance and dyspnea sensation.

Muscle function depends primarily on perfusion, muscle mass, fiber composition, and energy metabolism [1]. However, clinically, measurements of pressure, electromyography, and motor unit firing rates for ventilatory muscles [7], and strength and endurance for locomotor muscles [12], muscle oxygenation and perfusion measurements may be alternative methods to assess muscle function [13,14,15,16,17], and they have been reported to potentially be superior to the transdiaphragmatic tension time index [18]. Combining the indocyanine green dye (IGD) technique and near infrared spectroscopy (NIRS) has been reported to improve measurements of the peripheral muscle blood flow in healthy subjects and patients with COPD [19,20,21,22]. However, the IGD technique is sophisticated and its invasiveness related to the need to obtain arterial blood samples means that it is not widely used clinically. Subsequently, a relatively minimally invasive modified NIRS and IGD method to measure blood flow index was developed. The blood flow index has been shown to be highly correlated with muscle blood flow in ventilatory muscles and leg muscles in healthy subjects and patients with COPD [23,24,25]. However, the absolute blood flow cannot be determined using the blood flow index method.

With advances in NIRS technology, frequency-domain NIRS (fdNIRS) is capable of providing absolute values rather than relative changes in value [26]. This improvement allows for direct comparisons of oxygenation in different musculatures and different groups of subjects. Therefore, using fdNIRS, this study aimed to investigate differences in the balance of offering and utilizing oxygenation in leg and ventilatory musculature in non-obese healthy subjects and patients with COPD and CHF. This is the first study to compare the oxygenation of ventilatory and leg muscles between patients with COPD and CHF who have similar daily physical activity.

## 2. Methods

### 2.1. Study Design

In this observational study, we compared oxygenation and perfusion in two local musculatures during exercise as measured by near infrared spectroscopy (NIRS) across three groups of participants (healthy subjects, patients with COPD, and patients with CHF). The local Institutional Review Boards of Chung Shan Medical University Hospital (CS11144 and CMRPG32058) approved this study, which was conducted in compliance with the Declaration of Helsinki. Signed informed consent was obtained from each participant. The study was registered at Chung Shan Medical University Hospital (CSH-2012-C-23).

### 2.2. Subjects

Adult subjects were screened, enrolled, and allocated into COPD, CHF, and control groups from two university hospitals in Taiwan if they agreed to participate in the study. COPD was defined according to the GOLD guideline as follows [27]: a history of cigarette smoking for ≥10 pack-years; forced expired volume in one second and forced expired vital capacity ratio (FEV_1_/FVC) ≤0.7, and no significant bronchodilation in a stable condition. Stable CHF was defined as stable heart failure with a mid-range or reduced ejection fraction (EFmrHF or EFrHF, i.e., EF ≤ 49%) with or without peripheral vascular disease as confirmed by the cardiologists of the institutes [28]. Healthy subjects were defined as those with no obvious history of chronic disease such as severe hypertension, uncontrolled mild hypertension [29], other cardiovascular diseases, diabetes mellitus, organ disease, autoimmune diseases, cancer, anemia, psychiatric diseases or aliments of the lower extremities affecting exercise, and acute illnesses.

### 2.3. Measurements

#### 2.3.1. Clinical Assessment Tests

The subjects used the oxygen-cost diagram (OCD) [30] to assess daily activities. They were asked to indicate a point on an OCD, a 100-mm-long vertical line with everyday activities listed alongside the line, above which breathlessness limited them. The distance from zero was measured and scored. The New York Heart Association functional classification (NYHAfc) and modified Medical Research Council (mMRC) scales were also used to grade the functional status of the cardiopulmonary participants [28,30]. The mMRC scale includes 5 grades from 0 indicating “I only get breathless with strenuous exercise” to 4 “I am too breathless to leave the house”.

#### 2.3.2. Pulmonary Function Testing

FEV_1_, total lung capacity (TLC), and diffusing capacity for carbon monoxide (D_L_CO) were measured using spirometry and body plethysmography in accordance with the currently recommended standards.

#### 2.3.3. Cardiopulmonary Exercise Testing (CPET)

Each subject completed a ramp-pattern exercise test from resting to unloading, and the limit of their tolerance. Work rate was selected at a rate of 5–20 watts/minute depending on their fitness so that they could complete the loaded exercise within 8–12 min. Their fitness was evaluated using an OCD scale, and the ramp slope was selected on the basis of their fitness [31]. The time for loaded exercise was 8.4 ± 1.7 min for all subjects. Oxygen uptake ($\dot{V}$O_2_, mL/min), CO_2_ output ($\dot{V}$CO_2_, mL/min), minute ventilation ($\dot{V}$_E_), blood pressure, oxyhemoglobin saturation assessed by pulse oximetry (S_P_O_2_), blood pressure, and Borg dyspnea score were measured.

#### 2.3.4. Near Infrared Spectroscopy (NIRS)

An OxiplexTS^TM^ system (ISS, Champaign, IL, USA) was used to measure deoxygenated and oxygenated hemoglobin and intracellular myoglobin (HHb+HMb and HbO_2_+MbO_2_) [32,33]. Even though (Mb) is likely to contribute 60%–90% to the NIRS signals coming from skeletal muscles [34], for simplicity, we used HHb and HbO_2_ instead of HHb+HMb and HbO_2_+MbO_2_. The distance between the transmitter and receiver optodes of the NIRS system was 4 cm. NIRS is usually expressed by relative values instead of absolute values. However, the OxiplexTS^TM^ system can provide absolute values of tissue saturation in real time [26,35], and it is therefore possible to explicitly compare the magnitudes of responses in tissue HHb, HbO_2_, the sum of both (total blood capacity, TOT), and tissue saturation or the tissue oxygenation index (the fraction of oxygenation and sum of the two laser diode wavelengths, S_m_O_2_, %). As TOT cannot be used to represent blood flow [36], we defined the sum of HHb and HbO_2_ as the total blood capacity, which included arterial, capillary, and venous blood volumes. A calibration block was used before each test as recommended by the manufacturer. The leg probe of the OxiplexTS^TM^ system was positioned over the vastus lateralis muscle 10–12 cm above the knee, parallel to the major axis of the thigh [37], and the thoracic probe was positioned approximately at the sixth intercostal space along the anterior axillary line [38] to monitor the anterior serratus and intercostal muscles [39]. We selected the anterior serratus muscle because fatigue of this muscle is accompanied with diaphragmatic fatigue during incremental exercise [40] and during the latter stages of maximal exercise (75% of maximum $\dot{V}$O_2_ and more [41]), and becomes dominant in hyperventilation [42,43]. A piece of plastic wrap was used to prevent the lamp from becoming fogged by sweat during exercise. An elastic strap was used to secure the leg probe and adhesive tape was used to secure the chest probe. The sampling frequency of the fdNIRS device was set at 1 Hz. For simplicity, NIRS data were averaged and presented for the last 15 s of rest, unloaded pedaling and peak loaded exercise. Of note, at peak exercise, the nadir was used for variables that decreased compared to the baseline and the peak was used for variables that increased. These data were used for information on the dynamic balance between O2 supply (possibly also including O2 store with myoglobin and cytochrome c) and O2 demand. We did not record the respiratory compensation point or the anaerobic threshold, as they are difficult to identify and measure at a single location during incremental exercise [44,45].

### 2.4. Statistical Analysis

Data were summarized as mean ± standard deviation. The sample size was estimated to be at least 10 for each group when the population mean difference in S_m_O_2_ was 5.5 with a standard deviation for the normal and CHF groups of 4 and with a significance level of 0.05 and a power of 0.8 [46]. For each outcome variable, the comparisons were planned a priori. In the univariate analysis, *p* values were calculated by analysis of variance (ANOVA) with Tukey’s correction for multiple comparisons to compare means across the three groups. The paired *t* test was used to compare two related means between different time points. Fisher’s exact method was used in a contingency table analysis for categorical variables. Muscle mass is related to age, body height and weight [47,48,49], and muscle hypoxia is affected by hypoxemia. Thus, general linear models were used in comparison of the three groups (normal, COPD, CHF), adjusting for potential confounders such as age, body mass index, and SpO2 at each time point, respectively. For comparisons within different time points for the same variable, we did not adjust for other confounders as the measurements were from the same subject and the comparisons were planned a priori [5]. During loaded exercise, the HHb level is workload-related [35,42,50] and thus, the Hb variables were additionally corrected for peak workload. Pearson’s correlation coefficients were used to quantify the pair-wise relationships between NIRS data and the demographic, lung function and peak exercise variables. All statistical analyses were performed using SAS statistical software (9.4, SAS Institute Inc., Cary, NC, USA). Statistical significance was set at *p* < 0.05.

## 3. Results

### 3.1. Population Characteristics

A total of 175 subjects were screened and 134 subjects were retained for analysis after excluding 41 subjects (Figure 1 and Table 1). The subjects were allocated into three groups (Table 1). Differences in OCD, NYHAfc, and mMRC scores between the patients with COPD and CHF were insignificant (all *p* = NS). The etiologies of CHF are shown in Table 1. Differences in demographic data, symptom scores, resting lung function, exercise physiology, S_P_O_2_ and performance of peak exercise in the three groups are shown in Table 1 and Table 2.

### 3.2. Contrast of Leg Muscles to Ventilatory Muscles

In comparison to the leg muscles of all participants, the ventilatory muscles usually had lower HHb, HbO_2_, and TOT and higher S_m_O_2_ values at rest and during unloaded and loaded exercise (Figure 2). The lower value of S_m_O_2_ in the ventilatory muscles than in the leg muscles during loaded exercise only occurred in the CHF group (please see below and Figure 3).

### 3.3. Leg Muscles in the Three Groups

To better visualize the dynamic changes in oxygenation of leg and ventilatory musculatures for comparisons across the three groups, the same dataset as Figure 2 was used (Figure 3). At rest, the healthy group tended to have higher HbO_2_ and significantly higher S_m_O_2_ (*p* < 0.0001), whereas the COPD group tended to have higher HHb (*p* < 0.1) and the CHF group tended to have lower HbO_2_ (*p* < 0.1) and TOT (Figure 3, left panels). Across the three groups, scenarios in the Hb variables during unloaded and loaded exercise were the same as at rest. Unloaded exercise did not change HbO_2_ but decreased HHb, thereby decreasing TOT and increasing S_m_O_2_ in all subjects (Figure 3). Although unloaded exercise changed the four Hb variables, the directions and levels of change were similar across the three groups. Loaded exercise significantly increased HHb in all subjects (all *p* < 0.0001) but did not significantly change HbO_2_, thereby significantly increasing TOT and decreasing S_m_O_2_ (Figure 3). Although loaded exercise changed the four Hb variables, the directions and levels of change were similar across the three groups.

### 3.4. Ventilatory Muscles in the Three Groups

At rest, the healthy subjects had modestly higher or tended to have higher HbO_2_, TOT, and S_m_O_2_ (Figure 3, right panels, *p* < 0.1). Across the three groups, differences in the Hb variables during unloaded and loaded exercise were the same as at rest. Unloaded exercise did not change the four variables for all subjects, except that TOT and S_m_O_2_ were slightly increased in the COPD group (Figure 3). Loaded exercise significantly increased HHb (except the CHF group) and decreased HbO_2_ (all *p* < 0.05–0.0001), thereby slightly decreasing TOT (*p* < 0.1–0.001) and significantly decreasing S_m_O_2_ (*p* < 0.05–0.0001). The CHF group had significantly lower S_m_O_2_ than the COPD group at peak exercise (*p* < 0.01).

### 3.5. S_m_O_2_ Data at Peak Exercise versus Demographic, Lung Function and CPET Data

Nadir S_m_O_2_ levels obtained from the two musculatures had the best correlations with the demographic, lung function and CPET data at peak exercise compared to the other three Hb variables. For simplicity, only S_m_O_2_ was reported. Nadir S_m_O_2_ of the leg muscles was related to age (*r* = −0.36, Table 3), OCD (*r* = 0.30), maximal inspiratory pressure (MIP) %pred (*r* = 0.47), and some peak exercise variables, including heart rate (*r* = 0.30), $\dot{V}$_E_/maximal voluntary ventilation (MVV) ratio (*r* = −0.30), Borg score (*r* = −0.49), S_P_O_2_ (*r* = 0.37), and nadir $\dot{V}$_E_/$\dot{V}$CO_2_ (*r* = −0.30). Nadir S_m_O_2_ of the ventilatory muscles was related to age (*r* = −0.30), OCD (*r* = 0.29), cigarette smoking (*r* = −0.26), MIP %pred (*r* = 0.56), and some peak exercise variables including heart rate (*r* = 0.29) and $\dot{V}$_E_/MVV ratio (*r* = −0.34).

## 4. Discussion

The important findings of this study are that the leg muscles of all participants had higher HHb, HbO_2_ and TOT but lower S_m_O_2_ than the ventilatory muscles throughout the tests. In addition, the healthy controls had higher levels of S_m_O_2_ and HbO_2_ in both muscles throughout the tests, whereas the COPD group had higher HHb levels and the CHF group had lower HbO_2_ levels, which is consistent with a previous report [51]. Unloaded exercise changed the four Hb variables of the leg muscles more prominently than those of the ventilatory muscles. Loaded exercise increased HHb but did not change HbO_2_, thereby increasing TOT and decreasing S_m_O_2_ in the leg muscles. However, loaded exercise conversely decreased HbO_2_ and TOT in the ventilatory muscles and decreased S_m_O_2_ to a lower level in the diseased subjects than in the healthy subjects (Figure 3, *p* < 0.01) and to a further lower level in the CHF group (*p* < 0.01). The correlation study revealed some factors that were detrimental and others beneficial to SmO_2_ and by interventions, the SmO_2_ might be improved (please see Section 4.5. Correlations). The findings of this study highlight the feasibility of using fdNIRS and the importance of measurements of muscle oxygenation and blood volume changes in addition to clinical assessment tools for patients with exertional dyspnea, and thus offer a critical reference for clinicians in the management of such patients with ventilatory muscle weakness as ventilatory muscle weakness may aggravate ventilation-perfusion mismatch during exercise [52].

### 4.1. Re-Definition of HHb, HbO_2_, TOT, and S_m_O_2_

Changes in HHb and HbO_2_ were reported to indicate oxidative metabolism [5], i.e., that HHb represents tissue oxygen consumption and HbO_2_ represents tissue oxygen perfusion or delivery, respectively. TOT was reported to indicate recruitment strategies of circulation [5], also called O_2_ diffusion capacity [44,53] or to be related to vessel conductance [54] and S_m_O_2_ was reported to indicate the balance between muscle oxygen supply and demand [55].

However, muscle metabolism or blood flow cannot be evaluated using NIRS without performing arterial or venous occlusion [17,56,57,58] because HbO_2_ and HHb levels reflect the instantaneous balance between muscular oxygen consumption and delivery and probably need to take into consideration O_2_ store in the tissue of interest. During unloaded leg exercise, the leg muscles are the most active muscles (at least compared to ventilatory muscles), despite the workload being “~zero” watts, and pump HHb blood into veins and thus decrease HHb but preserve HbO_2_ (due to “~zero” watts, i.e., very light workload), thereby decreasing TOT and increasing S_m_O_2_. Oxygen consumption and total blood flow during this time cannot be lower than those at rest, so that HHb is not only an indicator of oxygen consumption but also an indicator of venous blood that is pumped away (i.e., decrease in HHb signal) or stored if any (i.e., increase in HHb signal) and therefore, TOT cannot represent blood flow but blood capacity (or volumes) of the tissue of interest [36,44,53,54]. Mancini et al. [38] and Terakado et al. [38,59] used isosbestic points of 800 nm and 805 nm to indicate blood volume rather than using TOT. Hirai et al. defined O_2_ diffusion as “flux of O_2_ driven by O_2_ pressure differential between the capillary and the intramyocyte milieu operating against a finite O_2_ diffusing capacity” [13], which is quite different from the definitions reported in prior studies [44,53]. The difference between HbO_2_ and HHb represents the balance between oxygen supply and extraction, and it probably resembles the meaning of S_m_O_2_, although not mathematically [16,58]. Of note, Legrand et al. used the difference in the absorption between 730 and 850 nm to indicate muscle oxygenation [14], whereas Mancini et al. used the difference in absorption between 800 and 760 nm for hemoglobin oxygen saturation [38], Mancini et al. and Chuang et al. used the difference in absorption between 760 and 850 nm for muscle deoxygenation [18,37], and Terkado et al. and Watanabe et al. used HbO_2_ = x_1_ΔA780 + y_1_ΔA805 + z_1_ΔA830 and HHb = x_2_ΔA780 − y_2_ΔA805 − z_2_ΔA830 (x_1,2_, y_1,2_, z_1,2_, coefficients; ΔA, change in the absorbance) [54,59]. In addition, Lucero et al. reported HbO_2_ and HHb values without providing the formula using the wavelengths [57].

### 4.2. Leg Muscles Versus Ventilatory Muscles

Blood volume has been reported to be higher in leg muscles than in ventilatory muscles [17,59]. This may have caused a higher blood volume per unit of tissue of interest in the leg muscles than in the ventilatory muscles and thus, the systemically higher values of HbO_2_ and HHb and TOT in the leg muscles compared to the ventilatory muscles, which is compatible with a previous study [17]. However, S_m_O_2_ was higher in the ventilatory muscles than in the leg muscles in this study. We speculate that the high S_m_O_2_ in the ventilatory muscles may be related to type I/II ratio, capillary/muscle density or capillary/myofibril ratio. Aerobic muscle enzyme activities have been reported to be higher in the ventilatory muscles in patients with COPD and CHF, whereas some anaerobic muscle enzyme activities have been reported to be higher in the quadriceps femoris [1]. In contrast to the leg muscles, the ventilatory muscles do not change much during unloaded exercise as they are involved only when exercise progresses to >85% of peak $\dot{V}$O_2_ [44]. This might be different from the sternocleidomastoid muscle which usually demonstrates increased activity at 30% of maximal inspiratory pressure in patients with COPD [60].

### 4.3. Loaded Exercise-Leg Muscles

In the present study, the leg muscles had preserved HbO_2_ and increased TOT due to the increase in HHb in all participants during loaded exercise, which is consistent with a report by Vogiatzis et al., who used NIRS with the IGD method to study patients with COPD [22]. This suggests that the blood flow supply was adequate or maintained to an extent that oxidative metabolism or demand or consumption (i.e., HHb) increased to a larger extent. However, Reid et al. reported that HbO_2_ decreased in biceps in response to an incremental forearm contraction in patients with COPD [16].

In the present study, HHb reached a similar level under maximal exercise across the three groups despite the different physiological limitations and exercise loads. This suggests that the peak capability of skeletal muscle oxygen extraction was similar across the three groups at peak exercise with the heart rate/predicted heart rate ratio reaching a similar level (Table 2, *p* = 0.11), and that a unique decrease in S_P_O_2_ may contribute to the mechanism in patients with COPD (Table 2, *p* = 0.009). This is compatible with a previous study by Lanfranconi et al., who reported that during incremental exercise, changes in HHb at peak exercise and $\dot{V}$O_2peak_ were lower in heart transplant recipients than in normal controls [49]. However, at lower exercise intensity in constant work rate exercise, they reported no difference in the kinetics of HHb between the two groups, despite the former having slower kinetics of $\dot{V}$O_2_ and heart rate [49]. The authors reported that the peak heart rate was 133.8 ± 3.8 b/min in patients aged 50.4 ± 2.6 years versus 173.0 ± 4.8 b/min in normal controls aged 47.3 ± 3.0 years; the heart rate stress level was estimated to be 79% versus 100%.

However, Katz et al. reported that in an invasive femoral venous blood study, patients with heart failure had higher oxygen extraction in leg muscles at peak exercise than healthy subjects [51]. Similarly, Zelt et al. reported that in patients with mild COPD, there was higher oxygen extraction in forearm muscles performing a continuous isometric handgrip squeeze at 20% of maximal voluntary contraction for 1.5 min compared to age-matched normal subjects [11]. We speculate that (1) the femoral vein and NIRS provide different information, as the former collects both superficial and deep leg veins whereas the latter collects the superficial microcirculation, and (2) all of the previous studies and the present study used different exercise protocols and different methods to calculate HbO_2_, HHb, and TOT, and measured different groups of muscles, and that all of these factors may have caused the difference in the results.

### 4.4. Loaded Exercise-Ventilatory Muscles

In the present study, in the ventilatory muscles, loaded exercise decreased both HbO_2_ and TOT unequally across the three groups, which is compatible with previous studies in patients with COPD and normal subjects [14,22], suggesting that blood flow to the ventilatory muscles did not match the need for oxidative metabolism. The healthy subjects had higher HbO_2_, TOT, and S_m_O_2_, whereas the CHF group had lower HbO_2_, TOT, and HHb, and the CHF group had the lowest S_m_O_2_ at peak exercise. We speculate that the CHF group had poor perfusion in the ventilatory muscles at peak exercise, which is compatible with previous reports [5,22,59]. Thus, this study does not support the metaboreflex or steal phenomenon of blood redistribution during heavy exercise [14,20,21,22,38,59]. This phenomenon has also not been supported in previous reports, as both blood volumes of ventilatory muscles and leg muscles decrease from the breakpoint of muscle oxygenation (i.e., ~74–88% of $\dot{V}$O_2peak_) to peak exercise [14,50]. In addition, molecular mechanisms of diaphragm muscle atrophy in patients with severe COPD have been reported to be downregulated in muscle-specific microRNA expressions and to be higher in histone deacetylase 4 and myocyte enhancer factor 2C protein levels [61]. Further studies are needed to investigate whether these findings are related to impaired peripheral oxygenation caused by chronic poor perfusion in ventilatory and leg muscles and whether they can be extrapolated to patients with CHF. In addition, molecular and NIRS studies on the mechanisms of muscle atrophy in both ventilatory and leg muscles are warranted.

### 4.5. Correlations

In the correlation study, aging, high ventilation demand/capacity ratio, exercise hyperventilation, and high dyspnea rating were detrimental to S_m_O_2_, whereas inspiratory muscle strength and high heart rate and S_P_O_2_ at peak exercise were beneficial to S_m_O_2_. Thus, reducing ventilation demand/capacity ratio, exercise hyperventilation, and high dyspnea rating with interventions such as exercise training may improve S_m_O_2_. On the other hand, improving inspiratory muscle strength and S_p_O_2_ and enhancing the capacity of heart rate increase by exercise training and breathing maneuvers with O_2_ breathing may also improve S_m_O_2_. Lastly, $\dot{V}$O_2peak_ %predicted and peak work rate %predicted were not related to any of the four Hb variables of both musculatures in the present study (Table 3), compatible with a previous study [54]. This indicates that oxygen extraction (HHb) and the other three variables were not related to exercise capacity. However, changes in HHb at peak exercise in the leg muscles have been reported to be highly related to $\dot{V}$O_2peak_ (*r* = 0.9, *p* < 0.0001) in patients with mitochondrial myopathy and McArdle’s disease [48]. In patients with mitochondrial myopathy and McArdle’s disease, myofibrils extract approximately 20% of the oxygen, whereas in normal controls, they extract approximately 70% [48]. In patients with heart failure or heart transplantation, myofibrils extract normal [49] or even higher amounts of oxygen [51].

### 4.6. Study Limitations

Age and body mass index were different across the three groups and thus, selection bias may have been introduced. The issue of choosing different exercise increments among the subjects during CPET could have affected the pattern of oxygenation/deoxygenation responses between the three groups and should be considered to be a limitation of the study. Combining the IGD technique and NIRS can improve measurements of the peripheral circulation [20,21,22]. Given that the present study cannot conclusively refute the steal phenomenon and that the IGD technique is sophisticated, studies using central hemodynamic response measurements with a thermodilution method combined with NIRS and IGD to investigate systemic cardiac output and local circulation of locomotor and ventilatory muscles during exercise are warranted to investigate the steal phenomenon [21,38,62]. Time intervals and levels of exercise intensity are also important when studying the steal phenomenon. Additionally, the potential mechanism(s) orchestrating the study results cannot be identified in the present study because cardiac output, systemic oxygen delivery, leg and ventilatory muscle blood flow and oxygen delivery, muscle mass morphological and structural assessments were not evaluated. Moreover, when a single probe is used on large muscles, the NIRS signals may not represent the whole musculature, as potential heterogeneity of tissue responses limits its capability [34]. The thickness of subcutaneous fat was not measured in the present study because it has been reported to be a minor factor as it is limited to <10 mm [63]. For direct comparisons of HHb, HbO_2_, and TOT between individuals when using fdNIRS, correcting for adipose tissue thickness (ATT) for these variables has been recommended [34]. In our previous report, the triceps skinfold thickness was 5.5–6.7 mm in patients with COPD with a body mass index ranging from 20.7 ± 2.3 to 22.7 ± 3.8 kg/m^2^ [64]. As thigh skinfold is correlated with body mass index [65], this issue is expected to be minor as most of our participants had a body mass index between 23.8 ± 3.0 and 26.3 ± 3.1 kg/m^2^. Previous studies have reported skinfolds at the sixth intercostal space and the anterior axillary line ranging from 6.7 ± 2.3 to 6.7 ± 4.6 mm, and a skinfold at the vastus lateralis muscle ranging from 8.6 ± 4.6 to 8.4 ± 2.9 mm in young subjects with a BMI of 21.7 to 23.7 kg/m^2^ [14,17], and 6.5 ± 1.0 mm in middle-aged normal subjects and heart transplant recipients with a BMI of 25.1 to 26.2 kg/m^2^ [49]. Although a reference was available to correct the NIRS signals after considering ATT, only seven male participants were included in their study, which raises concerns about its generalizability [66]. The cutaneous circulation may contribute to Hb NIRS signals in both muscles [67]; however, this contribution may be controversial. In normal subjects, at moderate intensity exercise and at maximal exercise, Anderson reported that total cutaneous blood flow decreased from 2 L/min to 1 L/min while total muscular blood flow increased from 12 L/min to 22 L/min [68]. To discriminate perfusive or diffusive defects as the mechanism of impaired oxygen transport [13], high-power time-resolved NIRS should be used. However, this equipment is still costly. Lastly, the sample size was small and further studies are needed with a larger number of participants. Despite these limitations, through careful interpretation and by not enrolling obese subjects, fdNIRS may still be useful to measure muscular metabolism and microcirculation.

## 5. Conclusions

This study differentiates the Hb oxygenation status of locomotor and ventilatory muscles between healthy and diseased subjects using frequency domain NIRS which can yield absolute values. The status of Hb oxygenation was different in different musculatures, and changes in oxygenation and S_m_O_2_ in response to loaded exercise between locomotor and ventilatory muscles were related to the type of disease. When performing peak exercise, more attention should be paid to ventilatory muscles in patients with cardiopulmonary diseases, especially those with CHF.

## 6. Future Perspectives

To better understand the potential mechanisms orchestrating dynamic muscle oxygenation in different types of diseases, directly measuring cardiac output, systemic oxygen delivery, leg and ventilatory muscle blood flow and oxygen delivery, and muscle mass morphological and structural changes and molecular mechanisms of muscle atrophy concomitantly with tissue monitoring with near infrared spectroscopy is warranted. This may be also suitable for interpreting the effects of pharmacological and non-pharmacological interventions in patients with exertional dyspnea.

## Figures and Tables

**Figure 1 jcm-08-01641-f001:**
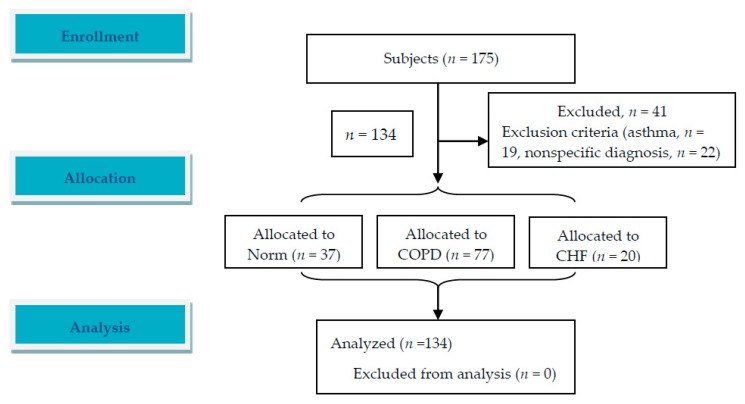
An image of flow diagram. A total of 175 subjects were screened and 134 patients were enrolled and analyzed. Follow-up was not applicable in this study.

**Figure 2 jcm-08-01641-f002:**
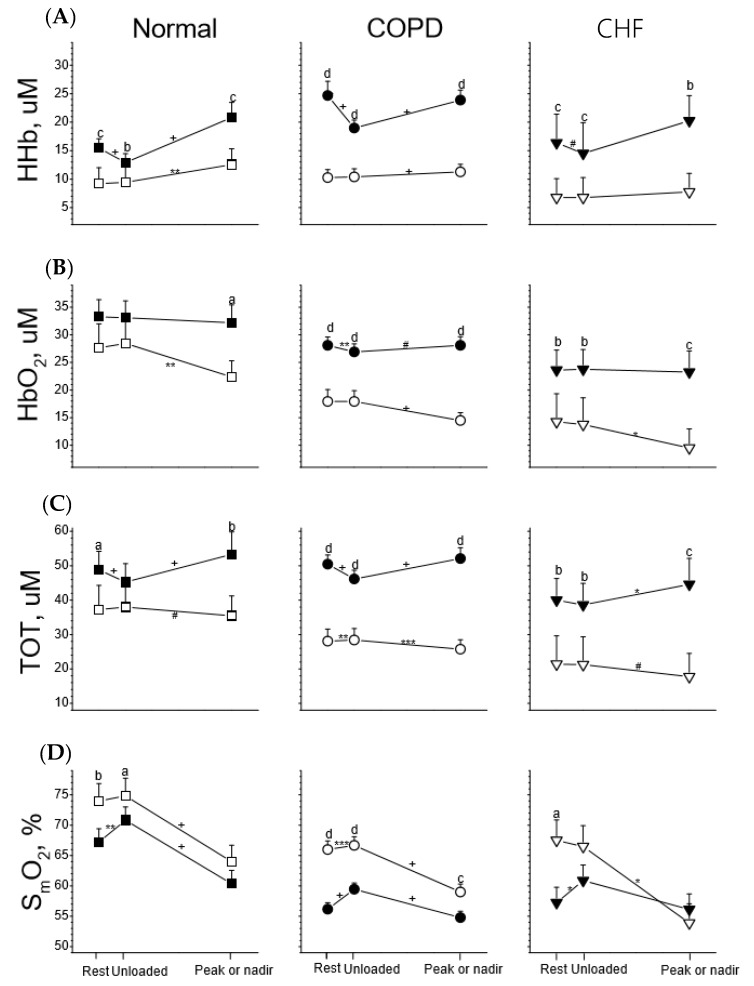
Tissue deoxygenation (HHb, row (**A**)), oxygenation (HbO_2_, row (**B**)), total blood capacity (TOT, row (**C**)), and saturation (S_m_O_2_, row (**D**)) of leg muscles (solid symbol) and ventilatory muscles (open symbol) in response to incremental exercise in healthy subjects (Normal) and subjects with chronic obstructive pulmonary disease (COPD) and congestive heart failure (CHF). Paired *t* tests were used for comparison between two muscles at three stages of the exercise test, respectively: ^a^
*p* < 0.05, ^b^
*p* < 0.01, ^c^
*p* < 0.001, ^d^
*p* < 0.0001; paired *t* tests were used for comparisons of changes in variables between rest and unloaded exercise and between unloaded exercise and at peak (or nadir) exercise, respectively: ^#^
*p* < 0.1, * *p* < 0.05, ** *p* < 0.01, *** *p* < 0.001, ^+^
*p* < 0.0001.

**Figure 3 jcm-08-01641-f003:**
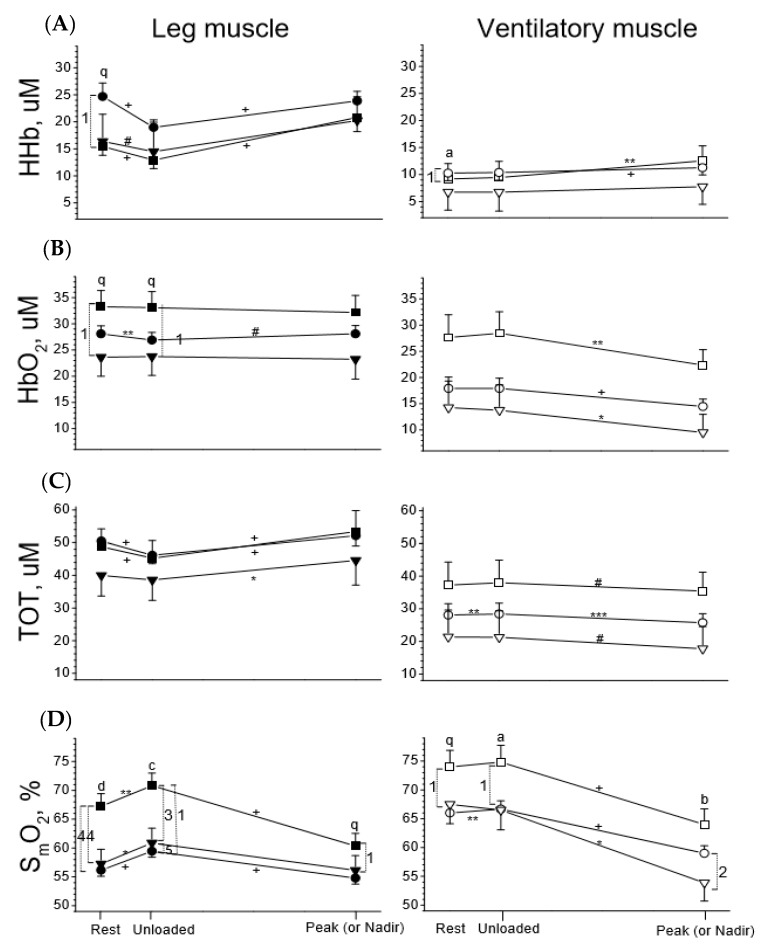
Tissue deoxygenation (HHb, row (**A**)), oxygenation (HbO_2_, row (**B**)), total blood capacity (TOT, row (**C**)), and saturation (S_m_O_2_, row (**D**)) of leg muscles (left panels, solid symbols) and ventilatory muscles (right panels, open symbols) in response to incremental exercise in healthy subjects (square symbol) and subjects with chronic obstructive pulmonary disease (COPD, circle symbol) and congestive heart failure (CHF, down triangle symbol). ANOVA was used for group comparisons at 3 stages of exercise, respectively: ^q^
*p* < 0.1, ^a^
*p* < 0.05, ^b^
*p* < 0.01, ^c^
*p* < 0.001, ^d^
*p* < 0.0001; Paired *t* tests were used for comparisons of changes in variables between rest and unloaded exercise and between unloaded exercise and at peak (or nadir) exercise, respectively: ^#^
*p* < 0.1, * *p* < 0.05, ** *p* < 0.01, *** *p* < 0.001, ^+^
*p* < 0.0001. Comparisons of variables between two groups of subjects, Arabic numbers ^5^
*p* < 0.1, ^1^
*p* < 0.05, ^2^
*p* < 0.01, ^3^
*p* < 0.001, ^4^
*p* < 0.0001.

**Table 1 jcm-08-01641-t001:** Demographic data, symptom scores, lung function and oxyhemoglobin saturation measured with pulse oximetry (S_P_O_2_) and oxygen uptake (VO_2_) at rest.

Group	Normal		COPD		CHF		ANOVA
	mean	SD	mean	SD	mean	SD	*p* Value
*n* = 134	37		77		20		
Age, years	54.5	17.5	66.4	11.1	52.1	12.6	<0.0001
Gender, M/F	33/4		70/7		19/1		0.76
Body mass index, kg/cm^2^	23.8	3.0	24.0	3.7	26.3	3.1	<0.05
Oxygen-cost diagram, cm	8.6	1.2	7.0	1.2	7.6	1	<0.0001
NYHAfc, I, II, III (%)	-		37.9, 51.7, 10.3		47.1, 52.9, 0		0.58
mMRC, 0, 1, 2 (%)	-		41.4, 48.3, 10.3		70.6, 29.4, 0		0.14
Cigarette, pack⋅year	4.8	17.5	58	33.2	31.8	26	<0.0001
Co-morbidity, Yes/No							
Hypertension, *n* =	4/36 *		18/73 *		11/20		<0.01
Diabetes mellitus, *n* =	1/36 *		6/73 *		1/20		0.57
Lung function and others							
FVC %predicted, %	96	12	76	21	88	14	<0.0001
FEV_1_%predicted, %	97	10	59	22	88	14	<0.0001
FEV_1_/FVC ratio, %	81	7	59	15	82	5	<0.0001
Ejection fraction ^†^, %	-	-	-	-	44	7	NA
$\dot{V}$O_2_, %predicted, %	17	5	22	8	18	6	<0.01
S_P_O_2_, %	97	1	96	3	97	1	<0.001

COPD: chronic obstructive pulmonary disease, CHF: congestive heart failure. FVC: forced expired capacity, NYHAfc: New York Heart Association functional classification, mMRC: modified Medical Research Council, FEV_1_: forced expired volume in one second, * missing one and four subjects, respectively. Participants with CHF included 1 subject with ventricular septal defect, 2 subjects waiting for heart transplantations, 3 subjects with decompensated cardiomyopathy, 13 subjects with coronary artery disease, and 1 subject with hypertensive cardiovascular disease. ^†^ Ejection fraction of the left ventricle using 2-dimensional echocardiography.

**Table 2 jcm-08-01641-t002:** Cardiopulmonary exercise test at peak exercise.

Group		Normal				COPD				CHF				ANOVA	
		mean		SD		mean		SD		mean		SD		*p* Value	
Workload, WR, watt		144		32		87		40		116		28		0.0002	
WR, watt% predicted, (%*p*), %		107		26		78		27		81		27		0.42	
Respiratory exchange ratio		1.11		0.16		1.11		0.16		1.17		0.13		0.62	
Oxygen uptake, $\dot{V}$O_2_ %*p*, %		81		8		68		22		60		16		0.07	
$\dot{V}$O_2_, mL/min/kg		30.4		4		17.6		6.3		18.4		2.6		<0.0001	
$\dot{V}$O_2__anaerobic threshold% max, %		48		8		49		9		37		11		0.07	
Cardiac frequency, f_c_, %*p*, %		91		5		84		9		84		15		0.11	
$\dot{V}$O_2_/f_c_, mL/beat		11		2		9		3		8		2		0.09	
Systolic blood pressure, mm Hg		193		17		175		22		150		30		0.008	
Diastolic blood pressure, mm Hg		80		5		86		12		87		21		0.82	
Minute ventilation, $\dot{V}$_E_, L/min		59.4		12.5		43.7		17		61.7		18.9		0.002	
Breathing frequency, b/min		32		6		37		7		43		8		0.01	
Tidal volume, V_T_, L		1.88		0.36		0.98		0.45		1.32		0.36		<0.0001	
V_T peak_/vital capacity		0.5		0.06		0.47		0.1		0.45		0.09		0.46	
$\dot{V}$_E_/maximum voluntary ventilation		0.45		0.11		0.75		0.22		0.56		0.12		0.0001	
$\dot{V}$_E_/CO_2_ output		30.1		5		37.1		5.9		37.7		3.9		0.002	
S_P_O_2_, %		94.6		4		92.9		4.5		98.3		1.1		0.009	
Borg dyspnea at peak		NA		NA		6.1		1.5		4.8		1.3		0.05	

COPD: chronic obstructive pulmonary disease, CHF: congestive heart failure, S_P_O_2_: oxyhemoglobin saturation measured with pulse oximetry.

**Table 3 jcm-08-01641-t003:** Correlation coefficient (*r*) of oxygenation saturation of muscles at peak exercise with demographics, lung function and peak/nadir exercise in all subjects.

Demographics	Vastus Lateralis (*r*)	Serratus Anterior (*r*)
Age	−**0.36 ****	−**0.30 ***
OCD	**0.30 ***	**0.29 ***
Cigarette	−0.13	−**0.26 ***
BMI	0.21 ^¶^	0.23 ^¶^
Lung function		
MIP%	**0.47** ^†^	**0.56** ^††^
MEP%	0.07	0.14
FVC%	−0.03	−0.13
FEV_1_%	0.12	0.11
FEV_1_/FVC	0.24 ^¶^	0.11
Exercise		
HR	**0.30 ***	**0.29 ***
Watt%	−0.02	0.02
VO_2_%	−0.08	0.03
VO_2_/HR	0.01	0.13
dVO_2_/dWR	0.07	0.05
V_E_%	−**0.33 ****	−**0.34 ****
V_E_/VCO_2_	−**0.30 ***	−0.21
Borg	−**0.49** ^†^	−0.15
S_P_O_2_	**0.37 ****	0.04
V_T_/VC	−0.09	−0.13
B_f_	0.03	−0.08

OCD: oxygen-cost diagram, Cigarette: in pack × years, BMI: body mass index, MIP: maximal inspiratory pressure, MEP: maximal expiratory pressure, FVC: forced vital capacity, FEV_1_: forced expired volume in one second, HR: heart rate, d: slope, VO_2_: oxygen uptake, V_E_: minute ventilation, WR: work rate, VCO_2_: ratio of V_E_ and carbon dioxide output, S_P_O_2_: oxyhemoglobin measured by pulse oximetry, V_T_/VC: ratio of tidal volume and vital capacity, B_f_: breathing frequency. The numbers in bold indicate *p* < 0.05. ^¶^ 0.05 < *p* < 0.1, * *p* < 0.05, ** *p* < 0.01, ^†^
*p* < 0.001, ^††^
*p* < 0.0001.

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
