# Peer review of "More Impaired Dynamic Ventilatory Muscle Oxygenation in Congestive Heart Failure than in Chronic Obstructive Pulmonary Disease"

_jcm, 2019, doi:10.3390/jcm8101641_

Round 1

Reviewer 1 Report

I carefully read the resubmitted version of the manuscript. Authors have cautiously addressed my points raised previously in my report. The manuscript has been significantly improved compared to the original version and the innovation of the study is now highlighted. Please find a few minor points that need to be addressed by the authors. Therefore, I have no other comments to make but congratulates the authors for their hard work.

INTRODUCTION SECTION. Lines 36-37 Authors state ¨However, the IGD technique is sophisticated and not widely used clinically¨. Please use the following references and revise the introduction and discussion section accordingly. 1. Near-infrared spectroscopy using indocyanine green dye for minimally invasive measurement of respiratory and leg muscle blood flow in patients with COPD. J Appl Physiol (1985).125(3):947-959, 2018;  2. Blood flow index using
near-infrared spectroscopy and indocyanine green as a minimally invasive
tool to assess respiratory muscle blood flow in humans. Am J Physiol
Regul Integr Comp Physiol 300: R984 –R992, 2011;  3. Near-infrared
spectroscopy and indocyanine green derived blood flow index for noninvasive measurement of muscle perfusion during exercise. J Appl Physiol (1985) 108: 962–967, 2010.

INTRODUCTION SECTION. Last sentence of the introduction section (The findings of this.....mismatch during exercise) reads like a conclusion statement and should be moved into the discussion section.

METHODS SECTION. In my previous report, I highlighted the issue of choosing different exercise increments among subjects during CPET. According to this reviewer point of view, this could affect the pattern of oxygenation/deoxygenation responses between the three groups and should be considered as a limitation of the study.

METHODS SECTION. 2.3.4. Near-infrared spectroscopy (NIRS). Please provide information about the sampling frequency of the NIRS device. NIRS data were measured during the last 15 seconds of rest, unloaded pedalling and peak loaded exercise and authors present nadir values of all oxygenation variables. How nadir values were calculated? (i.e., was a single value of the 15 seconds of the recording period). How many values were included in these 15 seconds for further analysis? The nadir values for HBO2 HHB and SmO2 were always attained at the same time for all patients? Authors should give more details about this method of NIRS data analysis. 

Reviewer 2 Report

Dear authors,

thanks again for editing your manuscript.

However i still can not follow the adjustment method in the methods-section 2.4.

Did you adjust p-values or did you adjust the data itself?

Did you use any regression or linear models for this?

Please provide the exact method how you have adjusted the data!
